# The Enigma of Gender Differences in an Environment-Behavior-Health Model of Elderly People: The Choice Between Individually and Sociality

**DOI:** 10.3390/ijerph17103483

**Published:** 2020-05-16

**Authors:** Zhenhua Zheng, Junling Gao, Ye Yang

**Affiliations:** 1College of Architecture and Urban Planning, Tongji University, 1239 Siping Road, Shanghai 200092, China; 19527@tongji.edu.cn; 2School of Public Health, Fudan University, 220 Handan Road, Shanghai 200433, China; jlgao@fudan.edu.cn

**Keywords:** outdoor environment, behavior, elderly, health, gender difference, activity ability

## Abstract

Gender difference in the health of elderly people has become an important perspective to consider when studying their health. However, the gender differences in terms of “environment–behavior–health” from the social ecology perspective remain unknown. On the basis of survey data from Shanghai, China, including 2839 samples, this study explored the gender differences of outdoor environment–behavior–activity ability in elderly people from two aspects of the mean value of variables and the path relationship, utilizing the structural equation model analysis method combined with social ecology theory. The results showed that, at the level of the mean of variables, the elderly males had significantly higher mean values than the females in terms of individual behaviors, but had significantly lower values in terms of social behaviors. At the level of path relationships, there were similarities in that the health of elderly males was more likely to be affected by individual factors, whereas that of the females was more effected by social factors. Through the great positive effect of the high-level social behaviors, the health of the elderly females was progressively promoted. The study provided some useful insights for future research, aiming at promoting gender-sensitive urban design.

## 1. Introduction

The aging of the population is currently a problem worldwide. The health of elderly people not only affects the financial expenditure in various countries but also determines the health and quality of life of the world’s population. Gender differences have become an important issue in research that focuses on the problem of elderly people’s health [1]. There are two important aspects of these gender differences. One is the difference in their characteristics, as related research studies have shown that there is health inequity between male and female elderly people. The other is the difference in the influencing factors. Related research has contributed to the improvement in health inequity and social inequity. Moreover, in response to the different influencing factors, corresponding measures have been used to improve the health of both elderly male and female individuals, which will eventually promote the health of older people overall. 

Self-rated health has often been used in research as a measure of elderly people’s health [2,3], whereas objective evaluations, such as the ability for elderly people to perform particular activities, are seldom employed. In fact, the ability to perform activities determines the living conditions of elderly people and it affects their quality of life [4]. Through the objective evaluation of their activity ability, their living and health conditions can be effectively measured. Therefore, our study focused on the activity ability of elderly people, which was used to indicate their health status. 

However, no unanimous conclusion has been reached about gender differences in the activity ability of elderly people. Most research studies have concluded that women live longer than men, but that the women’s activity ability is worse and their disability rate is higher [5,6,7]. Yet, some scholars have pointed out that there are no significant gender differences in their activity ability [8,9]. 

Because it is practicable to intervene in and improve the outdoor environment and individual behavior, it is feasible to enhance the health of elderly people [10,11,12]. It has been widely confirmed that the environment has an important effect on the health of elderly people [13,14], but most attention has been paid to self-rated health, and there has been little research into the influence of the environment on elderly people’s activity ability. 

In the existing literature, scholars have analyzed the influences of different behaviors on the activity ability of elderly people, which mainly includes individual physical behavior [15,16,17], social interaction [18,19], and social participation [20,21]. However, there have been relatively few studies of the combined effect of the environment and behavior on their activity ability. Nevertheless, with the development of the concept of active aging and age-friendly communities, the need for an age-friendly environment for the community has been emphasized. The combined effect of the environment and behavior on elderly people’s health is highly valued by both governments and scholars. 

In recent years, with the development of research into the influence of the environment and behavior on elderly peoples’ health, the theory of social ecology has been increasingly applied in the research [22]. Policymakers and scholars from different disciplines have begun to pay attention to the combined effect of the environment and behavior on their health. The core point of social ecology theory is that the individual’s health is not affected by the environment and behavior independently, but simultaneously, and that these two factors are interdependent [23,24,25]. Furthermore, the mediating role of the individual’s behavior is emphasized in the environment’s effect on the individual’s health. In older people, the influence of the outdoor environment on their health is reflected in the effect on their behavior; that is, the environment influences their behavior, and the combination of the environment and their behavior has an influence on their health. 

Previous social ecology studies have often focused on the effect on elderly people’s health by examining the combined effect of the outdoor environment and one certain behavior [26,27]. However, the behavior of older people is multifaceted. The behaviors that have been widely confirmed to be helpful for elderly people’s health include outdoor leisure activities, neighborhood interaction, and social participation. These behaviors of older people represent different social levels. Outdoor leisure activities and social interaction separately reflect their behavior on the individual level and the social level, whereas social participation reflects a deeper social level. As these different levels of behavior have different effect mechanisms on elderly people’s health, it is scientifically valuable and instructive to integrate them into one model in order to conduct a comprehensive comparative study. 

Therefore, we aimed to explore the gender differences in elderly people’s health and the underlying mechanism using the social ecology framework [28,29], in order to come to a conclusion about two aspects: firstly, the characteristics of the gender differences in the behavior and health of elderly people, and secondly, the gender differences in the relationship between their outdoor environment, behavior, and activity ability. The conclusions will not only help us to understand the enigma of the gender differences in elderly people’s environment, behavior, and health, but they will also improve the health of both genders and enhance the health of older people overall. 

## 2. Research Hypothesis

Although there are few studies on the relationship between outdoor environment and the activity ability of the elderly, some scholars have confirmed that the built environment has a significant impact on elderly people’s activity ability. Chen and Wilmoth pointed out that a change in living environment affected the activity ability of older persons [4]. Kim indicated that the improvement of the living environment helped to enhance elderly people’s quality of life, as well as their activity ability [30]. Therefore, we proposed the following hypothesis: 

**Hypothesis** **1 (H1).**
*Good outdoor environment in the community has a significant positive impact on the activity ability of the elderly people.*


Most research studies of the influencing factors of elderly people’s activity ability have focused mainly on their individual behaviors, such as exercising outdoors, and on their social behavior, which is composed of social interactions and social participation. Outdoor exercise is considered a key intervention for improving the activity ability of elderly people [15,16,17,31,32,33], and gender differences in the habit of exercising outdoors are considered important factors in their health [34]. In addition, social behavior can significantly promote the activity ability of elderly people. For example, older people with active social interactions have better physical activity than those who are not socially active [18,19], and social participation can improve their activity ability [20,21]. It has also been found that social behavior has different effects on older people of different genders [35]. Therefore, we proposed the following hypotheses further: 

**Hypothesis** **2a (H2a).**
*The intensity increase of outdoor exercise could effectively enhance the activity ability of elderly people.*


**Hypothesis** **2b (H2b).**
*The intensity increase of neighborhood interaction could effectively enhance the activity ability of elderly people.*


**Hypothesis** **2c (H2c).**
*The intensity increase of social participation could effectively enhance the activity ability of elderly people.*


In terms of the theory of social ecology, the environment and the behavior do not affect people’s health independently, but rather interact with each other and play a part in people’s health at the same time [23,25,36], and it is emphasized that individual behavior plays a mediating role in the impact path from the environment to people’s health. Therefore, we considered that for elderly people, the community environment did not only impact the health of elderly people directly, but also intermediately, and the mediating effect worked through the paths of outdoor exercise, neighborhood interaction, and social participation. On the basis of the above analysis, we proposed the following hypotheses: 

**Hypothesis** **3a (H3a).**
*Outdoor exercise plays a mediating role in the impact path from the environment to people’s health.*


**Hypothesis** **3b (H3b).**
*Neighborhood interaction plays a mediating role in the impact path from the environment to people’s health.*


**Hypothesis** **3c (H3c).**
*Social participation plays a mediating role in the impact path from the environment to people’s health.*


On the basis of the above analysis, the hypothesis model “environment–behavior–activity ability” was proposed (Figure 1). 

## 3. Materials and Methods

The ethical approval code number is IRB#2015-12-0574 from the Institutional Review Board of the School of Public Health at Fudan University. The methods were carried out in accordance with the relevant guidelines and regulations, and written informed consent was obtained from all subjects. 

### 3.1. The Selection of the Communities and the Elderly Participants

In 2014, a household sample survey was conducted in Xinhua Street, Changning District, in the central area of Shanghai, China (Figure 2). It was performed to investigate the relationships between outdoor environment, outdoor exercise, neighborhood interaction, and social participation, and welfare of elderly people, such as their quality of life. A total of 2839 older people who were over 60 years old were included. 

Taking into account the geographical location, transportation convenience, and building construction time, 43 residential communities were selected from all of the communities in Xinhua Street. The diversity was considered so that the selected sample covered communities with different qualities (Figure 3). The older people over the age of 60 years in the 43 communities who volunteered to complete the survey were then selected as the target samples. They were given gifts by the investigators during the survey, and in order to control the validity of the questionnaire, the investigators cooperated with the elderly people during the whole process of the survey (answering the questions, etc.). After excluding surveys that were incomplete, there were 2783 final valid surveys, of which 1175 (42.2%) respondents were male and 1608 (57.8%) respondents were female. 

### 3.2. The Health and Activity Ability of Elderly People and Instrumental Activities of Daily Living

Due to its extensiveness, complexity, and individuality, it is difficult to measure the health of elderly people in the health research field. The assessment mainly includes measures of self-rated health, activity ability, chronic diseases, and frequency of seeing a doctor [2,3,37,38]. Of these, activity ability has an important effect on the living state and quality of life of older people [4]. In addition, scores of their activity ability can objectively reflect their physical function. 

As instrumental activities of daily living (IADL) are the basis of the relationship between elderly people and their surroundings [39], they can be used to measure the activity ability of elderly people. Different tools are used for specific research purposes, of which Lawton’s IADL scale is the most widely used [40,41]. It contains the eight indicators of shopping, outdoor exercise, cooking food, household chores, washing clothes, using a telephone, taking medications, and handling finances. 

### 3.3. Outdoor Environment

With the development of Post Occupation Evaluation (POE) theory [42], subjective assessment of the built environment has become one of the main methods of environment evaluation in recent years [43,44], which is the real response of the users to the environment. It can not only evaluate the quality of the existing environment, but also be used as a feedback mechanism to improve the environment. In addition, it helps to promote the theory and practice development of different disciplines, such as urban planning, and environmental and architectural design, which leads to the environment that serves users better. 

The measurement tool of a community’s perception of the environment that was developed by Mujahid et al. [45] was used to evaluate the outdoor environment. It consisted of the following seven dimensions: the convenience of walking, the walking environment, an adequate amount of trees, exercise opportunities, adequate sport facilities, the attractiveness of walking, and the attractiveness of exercise. All items were divided into five levels, and a higher score reflected the respondent’s greater acceptance of the community’s environment. 

### 3.4. Behavior of Elderly People

The assessment of behavior of elderly people consisted of two aspects: their individual behavior and their social behavior. As walking was the main form of exercise for the elderly people, it was used to measure their individual behavior. Two variables were examined: walking frequency and walking duration, wherein walking frequency measured the number of days every week when the walking duration was more than 10 min. 

Neighborhood interaction and social participation were the two aspects of older people’s social behavior that were assessed. Neighborhood interaction was measured using five indicators, which included mutual help, communication, chatting, caring, and activities. They were scored using a four-point scale that represented the frequency of neighborhood interaction in the community; that is, 1 (never), 2 (occasionally), 3 (sometimes), and 4 (often). 

Social participation was assessed in terms of the frequency that the elderly participants took part in social activities, which included voluntary work, self-management, mutual help activities, listening to lectures and presentations, participating in culture and sports activities, and attending interest groups. It was scored on a five-point scale that included 1 (never), 2 (several times a year), 3 (several times a month), 4 (once a week), and 5 (two to three times a week). A higher score represented a higher degree of social participation. 

### 3.5. Control Variables

Education, income, age, and time of living in the residential area were selected as the control variables in a structural equation model (SEM). Responses for each item of income ranged from 1 to 6 (1 = RMB < 1500, 2 = RMB 1500–2500, 3 = RMB 2500–3500, 4 = RMB 3500–4500, 5 = RMB 4500–5500, 6 = RMB > 5500), and the score from 1 to 5 (1 = middle school and below; 2 = high school, special school, or technical school; 3 = junior college; 4 = regular college; and 5 = master’s degree and above) represented education conditions. Time of living in the residential area meant the number of years living in this community. Self-rated health was also entered into the model as a control variable in order to avoid its influence on the relationship between the environment, behavior, and activity ability. 

### 3.6. Statistical Analyses

Descriptive statistical analysis and SEM were used in the study. By combining factor analysis with path analysis, SEM has the advantage in quantitative research of measuring multivariable interactions and group comparisons, which is helpful for identifying the logistic relationship between the quality of life, behavior, and outdoor environment of the elderly people. The maximum likelihood estimation method was employed, and the mediating effect was tested by referring to the method of MacKinnon et al. [46]. In order to examine whether the data were suitable for the SEM calculation, all variables were grouped using the 27nd and 73rd quartiles, and a *t*-test was performed. The results showed that all of the variables had good discrimination and were suitable for the SEM analysis. Because the sample number was as high as 2783 (>1000), the mean value of the samples had an approximately normal distribution, and the samples were suitable for the SEM analysis. 

The multifactor confirmatory analysis was performed on the five measurement models, which were the outdoor environment, outdoor exercise, neighborhood interaction, social participation, and activity ability. The reliability of the measured group was greater than 0.6, and the average variance extracted was above 0.5. The factor loadings of the observed variables were greater than 0.6, and the reliability coefficients (square multiple correlation) were above 0.36 [47]. As all of the measured models had good reliability and validity, they were suitable for the SEM analysis. The Cronbach’s alpha scores of the variables outdoor environment, outdoor exercise, neighborhood interaction, social participation, and activity ability were 0.900, 0.762, 0.868, 0.865, and 0.87, respectively. 

After the models were fit, the results showed that RMSER (root mean square error of approximation) met the ideal criterion, but other fit indexes did not; therefore, the model needed to be optimized. After the model optimization, the correlations between the residuals, such as e5 and e6, e9 and e10, e14 and e15, e15 and e16, e23 and e24, and e28 and e31, were established, and the final fit index met the criteria (i.e., *χ*^2^/*df* < 5, GFI (goodness of fit index) > 0.90, AGFI (adjusted goodness of fit index) > 0.90, CFI (comparative fit index) > 0.9, and root mean square error of approximation < 0.08), which meant that the model had a good fit (Table 1). 

## 4. Results

### 4.1. Gender Differences in the Variables

The descriptive statistics of the main variables are shown in Table 2. There were no significant differences in the mean values of all the observed variables between different genders. The mean value of walking duration was 28.61 and the walking frequency was 4.20 times per week, which indicated that the elderly people were used to walking. Both the walking frequency and walking duration of the elderly males were slightly higher than that of the elderly females. On the whole, the elderly people did not get used to neighborhood interaction or social participation, and the elderly females participated in more social activities than elderly males. As for the variables of the activity ability, the mean values of using a telephone, shopping, and handling finances were higher than others, and the abilities of washing clothes and cooking food were poor. The mean value of the age of elderly people was 71.35, and the mean values of education and income were both higher in elderly males than in elderly females. The elderly people had been living in the community for more than 21 years, which meant that the interviewees lived in a very familiar environment. The self-rated health (SRH) of the elderly people was close to the median, and that of the males was slightly higher than that of the females. 

### 4.2. Gender Differences in the Variables

On the whole, there were significant behavioral differences between the male and female elderly people, and the most significant difference was in social participation (Figure 4). The mean values of all of the five indicators (i.e., participating in community activities, participating in interest groups, volunteering, self-management, and listening to lectures) were significantly higher in the women than in the men, and the differences were 0.37 (*p* = 0.000), 0.32 (*p* = 0.000), 0.27 (*p* = 0.000), 0.26 (*p* = 0.000), and 0.20 (*p* = 0.000), respectively. In addition, there was also a significant gender difference in neighborhood interaction, such as chatting, gathering together, and communication, with the female participants again having significantly higher mean values than the males. The differences were 0.24 (*p* = 0.018), 0.21 (*p* = 0.034), and 0.20 (*p* = 0.036), respectively, for chatting, gathering together, and communication. Although the walking frequency of the elderly male participants was significantly higher than that of the elderly female participants, the difference was 0.14 days (*p* = 0.03). The control variables’ scores for as education, income, and self-rated health, were significantly higher in the elderly male participants than in the elderly female participants, and the differences were 0.67 (*p* = 0.000), 0.35 (*p* = 0.003), and 0.12 (*p* = 0.000), respectively. As for the IADL scale, for the behaviors of outdoor exercise, shopping, handling finances, and using the telephone, the scores of the elderly males were also significantly higher than those of the females, and the differences were 0.36 (*p* = 0.008), 0.09 (*p* = 0.000), 0.05 (*p* = 0.000), and 0.05 (*p* = 0.000), respectively. However, for the two activities of doing the laundry and cooking, the scores of the elderly male participants were significantly lower, and the differences were 0.18 (*p* = 0.000) and 0.17 (*p* = 0.000). In general, there were no obvious gender differences in the activity ability of the elderly population. 

### 4.3. Gender Differences in the Relationships between the Environment, Behavior, and Activity Ability

The group comparison results of the male and female elderly participants were analyzed, and it was found that there were significant differences between the males and females in the environment–behavior–activity ability model (Table 3 and Table 4, Figure 5 and Figure 6). 

For the elderly males, the outdoor environment had a significant positive effect on their activity ability, with a total effect of 0.100 (*p* = 0.011), and walking behavior had a significant positive effect on their activity ability, with a total effect of 0.232 (*p* = 0.001). However, neighborhood interaction and social participation did not have a significant effect on their activity ability. The control variables, such as age and self-rated health, also had a significant effect on their activity ability, with effect values of −0.145 (*p* = 0.005) and 0.293 (*p* = 0.000), respectively, whereas time of living in the residential area, income, and education had no significant effects. Therefore, hypotheses 1 and 2a were accepted for the elderly males, whereas Hypothesis 2b and 2c were rejected. 

The outdoor environment had a significant positive effect on walking behavior and neighborhood interaction for the elderly male participants, with values of 0.241 (*p* = 0.001) and 0.148 (*p* = 0.011), respectively, but it did not have a significant effect on social participation. Walking behavior had a significant positive affect on neighborhood interaction and social participation, with effect values of 0.165 (*p* = 0.005) and 0.123 (*p* = 0.040), respectively. 

The direct effect of the outdoor environment on the activity ability of the elderly male participants was not significant, but the indirect effect was significant, which indicated that there was a mediating variable in the path. This mediating variable was the walking behavior because the effects of neighborhood interaction and social participation on the elderly male participants were both insignificant. For the path from walking behavior to activity ability, walking behavior had a significant direct effect on the activity ability of the elderly male participants, whereas the indirect effect was not significant, which meant that there was no mediating variable in this path. The outdoor environment did not have a significant direct effect on neighborhood interaction, but the indirect effect was significant, which showed that walking behavior was also a mediating variable in the path between the outdoor environment and neighborhood interaction. Therefore, hypothesis 3a was accepted for the elderly males, whereas hypotheses 3b and 3c were rejected. 

For the elderly female participants, the outdoor environment did not have a significant effect on their activity ability, but walking behavior, neighborhood interaction, and social participation did have a significant positive effect, and their total effect values were 0.230 (*p* = 0.000), 0.165 (*p* = 0.000), and 0.183 (*p* = 0.000), respectively. The control variables of age, education, and self-rated health had significant effects on the activity ability of the elderly female participants, and the effect values were −0.106 (*p* = 0.006), 0.132 (*p* = 0.001), and 0.269 (*p* = 0.000), respectively. Time of living in the residential area and income did not have a significant effect on their activity ability. Therefore, hypotheses 2a, 2b, and 2c were accepted for the elderly females, whereas hypothesis 1 was rejected. 

The outdoor environment had a significant positive effect on the females’ walking behavior, and its total effect value was 0.164 (*p* = 0.005), but it had no significant effect on their neighborhood interaction and social participation. Walking behavior significantly affected neighborhood interaction positively, with an effect value of 0.160 (*p* = 0.002), but its effect on social participation was not significant. 

Walking behavior had both significant direct and indirect effects on the activity ability of the elderly female participants, which indicated that there was a partial mediating variable. Because the effect value of walking behavior on social participation was not significant, social participation was the partial mediating variable in the path from walking behavior to the activity ability of the elderly female participants. 

The outdoor environment had no significant impact on the elderly females, and thus was not possible that there was a mediating effect in this path. Therefore, hypotheses 3a, 3b, and 3c were rejected for the elderly females. The validation results of the hypotheses are shown in Table 5. 

## 5. Discussion

On the basis of the survey data of Shanghai and the theory of social ecology, by utilizing the environment–behavior–activity ability model, this paper discussed the gender differences in the health of elderly people from two levels, the mean of variables and the path relationships in the model. 

The gender differences in the health of the elderly people were not obvious. However, according to the environment–behavior–activity ability model, compared with the elderly female participants, the elderly male participants had significantly higher competence for some aspects in the level of mean of variables, such as walking frequency, income, education, and self-rated health, but a much lower engagement in neighborhood interaction and social participation. In particular, there was a significant gender difference in social participation, and all of the indicators of the elderly female participants’ social participation were significantly higher than those of the elderly male participants. This finding suggested that the aspects in which the elderly male participants had significant higher values reflected their individual characteristics; for example, socioeconomic status, individual exercise habits, and self-rated health. The aspects for which the elderly female participants had significantly higher values reflected the social characteristics, and social interaction and social participation were both social activities that these individuals took part in. 

Furthermore, at the level of path relationships in the model, there were significant differences between the male and female models, and they supported different items in our hypotheses. The male model validated the hypothesis that the outdoor environment has a significant positive impact on the activity ability of elderly males. However, the model came to a different result for females, that is, the outdoor environment did not impact the activity ability of the elderly females significantly. The female model validated the hypotheses that neighbor interaction and social participation impact elderly people’s activity ability significantly and positively, but the two hypotheses were rejected for the elderly males. 

On the whole, the gender differences were characterized by the fact that the elderly males were more easily influenced by the objective environment, whereas the elderly females were more influenced by social activities. 

Not only the activity ability of the elderly participants, but also the gender differences in the effect of the outdoor environment were reflected in their neighborhood interaction. The outdoor environment had a significant positive effect on the neighborhood interaction of the elderly male participants, but not the elderly female participants. 

However, there were also consistent results between the genders in the environment–behavior–activity ability model, which were mainly observed in the paths related to walking behavior, including its effect on activity ability and neighborhood interaction, as well as the effect of the outdoor environment. This result suggested that walking behavior had a significant positive effect on the activity ability and neighborhood interaction of the elderly people, and that it was stable and important. 

The environment influences behavior to a large extent [48,49,50,51], but in terms of our study, the outdoor environment had a stable significant effect only on the walking behavior of the elderly people. In addition, the significant effect of the outdoor environment on the neighborhood interaction was only found in the elderly males, and no significant effect on social participation was found in the elderly male or female participants. Thus, the behavior that has stronger individual features is much more likely to be affected by the outdoor environment, whereas behavior that has stronger social features is not likely to be affected. 

In the environment–behavior–activity ability model, the gender differences were shown in the mediating effect as well. The male model validated the hypothesis that walking behavior was the mediating variable in the impact path from outdoor environment to the activity ability, which was rejected by the model for the females. The results indicated that the outdoor environment affected the activity ability of the elderly male participants by affecting their behavior, which was the only path. Meanwhile, walking behavior was also the mediating variable in the path of the outdoor environment that affected the neighborhood interaction of the elderly male participants. Therefore, the effect of walking behavior on the elderly male participants’ activity ability was not only reflected by its direct effect, but also reflected in its mediating effect on the path from the outdoor environment to the activity ability and neighborhood interaction of the elderly male participants. 

In general, the gender differences in the environment–behavior–activity ability model had the following features: the elderly male participants were more influenced by individual and objective factors, whereas the elderly female participants were more influenced by social factors. According to the current features of the gender differences in the outdoor environment, the elderly females had significant higher values in social behaviors, such as neighborhood interaction and social participation; therefore, although the elderly females had significant lower values than the males in the individual factors, such as income, education, walking behavior, because of their higher participation in social behavior, their activity ability was not poor. Although the elderly males had higher socioeconomic status and performed better in outdoor exercise, the “individual” activities of the elderly males and the “social” activities explained the reason why there were no obvious gender differences in the activity ability of the elderly. 

The findings of the study are helpful for developing strategies to improve elderly people’s health, which is very important in the context of the aging population. Male and female elderly people should be considered separately so that more refined measures can be employed. For example, when community lectures, interest groups, or voluntary activities are conducted, more attention should be paid to female participants because these factors have a stronger effect on them. Furthermore, as the outdoor environment, especially the walking space, influences elderly male people to a great extent, their characteristics should be considered more during the design and construction of the environment. 

## 6. Conclusions

By analyzing the constructed environment–behavior–activity ability model, the features of the gender differences in the elderly people’s behavior and activity ability were explored. It was found that elderly male people were more likely to participate in individual activities, whereas elderly female people were more likely to take part in social activities and, correspondingly, it was found that the objective environment greatly affected the activity ability of the elderly male participants, whereas social activities greatly influenced the activity ability of the elderly female participants. On the basis of these findings, the outdoor environment should be constructed and built more specifically, taking into account the different uses in terms of gender; for instance, spaces for different activities should have different gender features in order to promote the comfort and health of the elderly population. 

However, there were limitations of this study. Firstly, as cross-sectional survey data were used in this study, the causal relationships between the outdoor environment, walking behavior, neighborhood interaction, social participation, and activity ability of the elderly people could not be fully explained, and a continuous longitudinal study should thus be conducted to support these findings in the future. Secondly, the evaluation of the elderly people’s health was based on the assessment of their activity ability. However, this was not comprehensive, and the assessment of more variables is required. In addition, the survey of the outdoor environment was based on a subjective assessment, and future studies should focus on combining the subjective perception and objective measurement, which would be helpful for identifying the differences and relationships between two models; that is, a “subjective perception environment–behavior–activity ability model” and an “objective measurement environment–behavior–activity ability model.” Finally, our survey data came out in 2014, and more than 5 years have passed since, and thus there may have been some changes in both the outdoor environment and in the elderly people’s health. Currently, we are conducting a new round of surveys, which will help to explore the relationship between the outdoor environment and elderly people’s health more deeply and continuously. 

## Figures and Tables

**Figure 1 ijerph-17-03483-f001:**
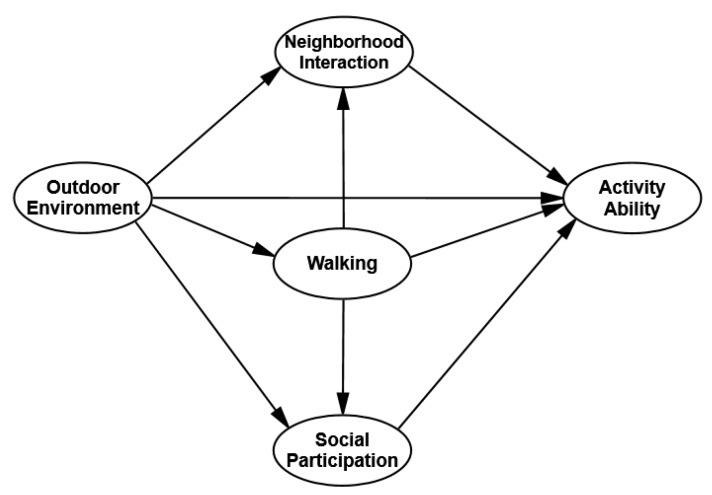
The hypothetical environment–behavior–activity ability model.

**Figure 2 ijerph-17-03483-f002:**
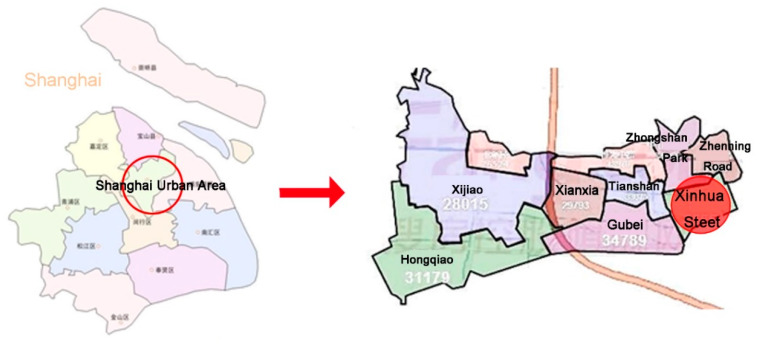
Location of Xinhua Street.

**Figure 3 ijerph-17-03483-f003:**
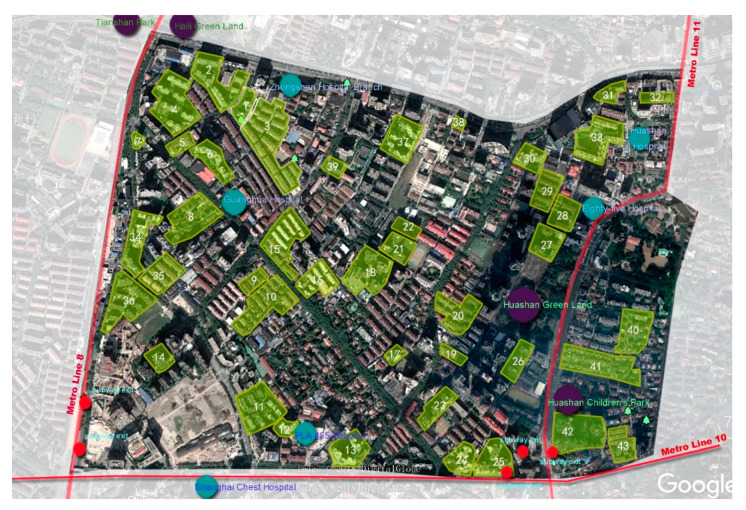
Map of the community sample (source: based on Google Maps, drawn by the author).

**Figure 4 ijerph-17-03483-f004:**
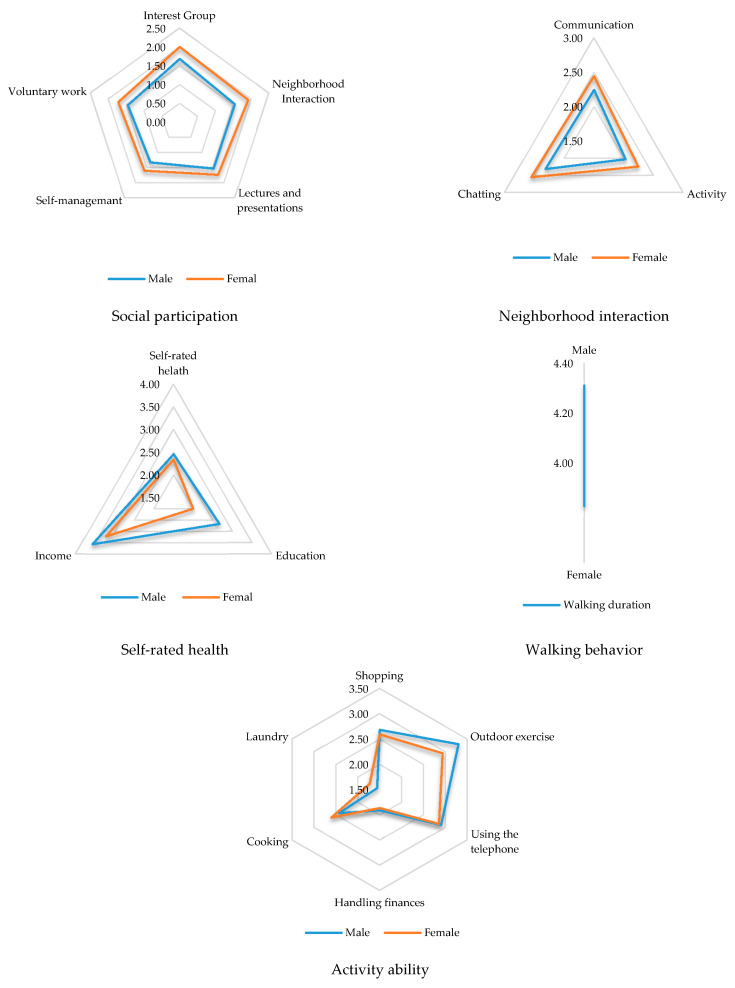
Gender difference of the behavior and health of the elderly population.

**Figure 5 ijerph-17-03483-f005:**
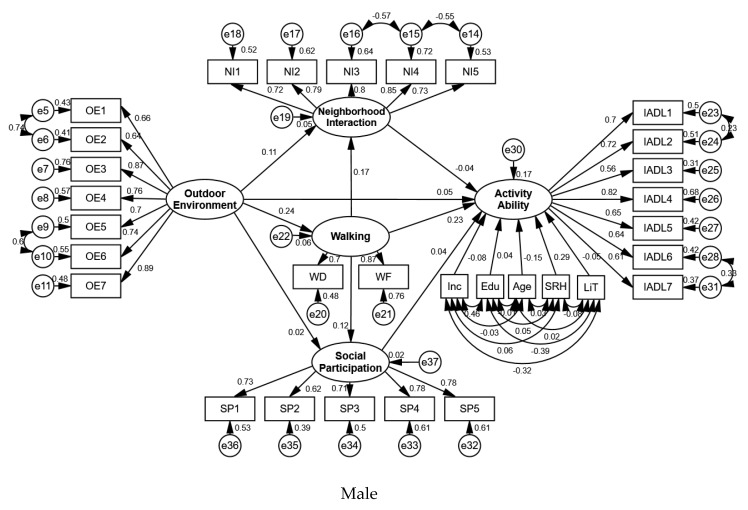
Standardization coefficients for male sample model.

**Figure 6 ijerph-17-03483-f006:**
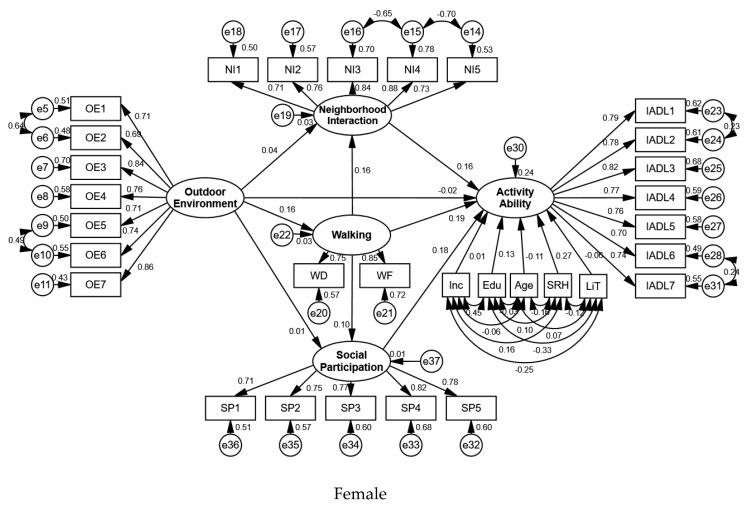
Standardization coefficients for female sample model.

**Table 1 ijerph-17-03483-t001:** Comparison of fit index before and after model optimization.

	GFI	AGFI	IFI	CFI	RMSEA	X^2^/DF
Pre-optimization model	0.819	0.798	0.843	0.842	0.059	6.237
Post-optimization model	0.907	0.904	0.925	0.924	0.037	2.568
Ideal standard	>0.9	>0.9	>0.9	>0.9	<0.08	<5

**Note:** GFI: goodness of fit index; AGFI: adjusted goodness of fit index; IFI: incremental fit index; CFI: comparative fit index; RMSEA: root mean square error of approximation; X^2^/DF: chi square degrees of freedom ratio.

**Table 2 ijerph-17-03483-t002:** Characteristics of variables.

Variable	Observed Indicators	Content of the Questions		Mean (All)	Mean (Men)	Mean (Women)
Outdoor environment	OB1: Exercise opportunities	Plenty of exercise opportunities in the community		3.00	2.99	3.00
	OB2: Sport facilities	Lots of sport facilities in the community		2.95	2.93	2.96
	OB3: Walking environment	Walking in the community is pleasant		3.29	3.26	3.30
	OB4: Adequate amount of trees	Trees in the community provide shadows		3.11	3.09	3.12
	OB5: Attractiveness of exercise	I often see other people doing exercise in the community		3.18	3.21	3.15
	OB6: Attractiveness of walking	I often see other people walking around the community		3.32	3.35	3.30
	OB7: Convenience of walking	It is very convenience to go out to many other places		3.64	3.62	3.65
Outdoor exercise	WF: Walking frequency	Walking frequency every week (at least 10 min)		4.20	4.31	4.17
	WD: Walking duration	How long is a walk (in minute)?		28.61	28.87	28.39
Neighborhood interaction	NI1: Mutual activities	How do you get together with each other or participate group activities in the community?		2.15	2.03	2.24
	NI2: Mutual chatting	How do you talk with each other on personal matters in the community?		2.34	2.45	2.44
	NI3: Mutual help	How do you help each other in the community, such as baby-sitting, shopping, etc.?		2.39	2.32	2.04
	NI4: Mutual communication	Communicating with neighbors at home		2.44	2.32	2.56
	NI5: Mutual caring	Taking care of neighbor’s house or belongings when he is not at home or traveling		1.91	1.86	1.94
Social participation	SP1: Voluntary work	Frequency of voluntary work		1.61	1.45	1.72
	SP2: Self-management	Frequency of attending the self-management or mutual help group		1.50	1.33	1.60
	SP3: Lectures and presentations	Frequency of listening to lectures and presentations		1.65	1.53	1.74
	SP4: Culture and sports activities	Frequency of attending culture and sports activities in the community		1.76	1.55	1.92
	SP5: Interest groups	Frequency of attending the interest groups		1.87	1.68	2.00
Activity ability	IADL1: Shopping	Range from 1 to 3	Conditions in the last month	2.63	2.68	2.59
	IADL2: Outdoor exercise	Range from 1 to 4		3.09	3.30	2.94
	IADL3: Cooking food	Range from 1 to 5		2.53	2.43	2.61
	IADL4: Household chores	Range from 1 to 4		3.26	3.31	3.22
	IADL5: Washing clothes	Range from 1 to 5		1.66	1.56	1.73
	IADL6: Using a telephone	Range from 1 to 3		2.88	2.90	2.85
	IADL7: Handling finances	Range from 1 to 4		1.88	1.91	1.86
Control variable	Age	How old are you?		71.35	72.31	70.58
	Inc: Income	Monthly income of your family		3.37	3.57	3.22
	Edu: Education	Your education condition		2.28	2.67	2.00
	LiT: Time of living in the residence	How long have you been living in this community?		21.66	22.09	21.35
	SRH: Self-rated health	Self-assessment of your overall health		2.38	2.45	2.34

**Table 3 ijerph-17-03483-t003:** Total effect, direct effect, and indirect effect in the model.

Groups	Independent Variables	Mediating Variables							Dependent Variables		
		Walking Behavior	Neighborhood Interaction			Social Participation			Activity Ability		
			Total Effect	Direct Effect	Indirect Effect	Total Effect	Direct Effect	Indirect Effect	Total Effect	Direct Effect	Indirect Effect
Male	Outdoor environment	0.241 ***	0.148 **	0.108	0.040 ***	0.053	0.023	0.030	0.100 **	0.047	0.053 ***
	Walking behavior	----	0.165 ***	0.165 ***	----	0.123 **	0.123 **	----	0.232 ***	0.233 ***	−0.001
	Neighborhood interaction	----	----	----	----	----	----	----	−0.039	−0.039	----
	Social participation	----	----	----	----	----	----	----	0.039	0.039	----
Female	Outdoor environment	0.164 ***	0.070	0.043	0.026	0.029	0.013	0.016	0.022	−0.025	0.047
	Walking behavior	----	0.160 ***	0.160 ***	----	0.098	0.098	----	0.230 ***	0.186 ***	0.044 ***
	Neighborhood interaction	----	----	----	----	----	----	----	0.165 ***	0.165 ***	----
	Social participation	----	----	----	----	----	----	----	0.183 ***	0.183 ***	----

Note: *** means significant at the 0.01 confidence level; ** means significant at the 0.05 confidence level.

**Table 4 ijerph-17-03483-t004:** Effect of control variables.

Groups	Control Variables	Dependent Variables (Activity Ability)
Male	Age	0.145 ***
	Income	−0.079
	Education	0.035
	Time of living in the residential area	−0.047
	Self-rated health	0.293 ***
Female	Age	0.106 ***
	Income	0.007
	Education	0.132 ***
	Time of living in the residential area	−0.053
	Self-rated health	0.269 ***

Note: *** means significant at the 0.01 confidence level.

**Table 5 ijerph-17-03483-t005:** The validation results of the hypotheses.

	Hypothesis	Results	
		Elderly Male	Elderly Female
1	1: Good outdoor environment in the community has a significant positive impact on the activity ability of elderly people.	√	X
2	2a: The intensity increase of outdoor exercise could effectively enhance the activity ability of elderly people.	√	√
	2b: The intensity increase of neighborhood interaction could effectively enhance the activity ability of elderly people.	X	√
	2c: The intensity increase of social participation could effectively enhance the activity ability of elderly people.	X	√
3	3a: Outdoor exercise plays a mediating role in the impact path from the environment to people’s health.	√	X
	3b: Neighborhood interaction plays a mediating role in the impact path from the environment to people’s health.	X	X
	3c: Social participation plays a mediating role in the impact path from the environment to people’s health.	X	X

Note: √: accepted; X: rejected.

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
