# Peer review of "The Enigma of Gender Differences in an Environment-Behavior-Health Model of Elderly People: The Choice Between Individually and Sociality"

_ijerph, 2020, doi:10.3390/ijerph17103483_

Round 1

Reviewer 1 Report

Lines 50-51 "It is easy to intervene....": In what sense it is easy? I believe it not easy at all!

Lines 53-54 Repetition, please erase.

Line 66 The "few studies" must be cited.

Line 72 The "theory of social ecology" must be cited.

Line 81 The "previous social ecology studies" must be cited.

Line 91 The "Social Ecology framework" must be cited.

Line 93 "Living environment" is a vague term, do you mean "Outdoor environment"? There are several scientific studies concentrating on Indoor living environment and human health.

Lines 94-96 Bold statements on own research should be avoided.

Line 124 Erase "and so on".

Lines 127-128 The sentence (and possibly the whole study, with this respect) is not clear, please explain better. Please note that IALS (i.e. cooking, cleaning, transportation, laundry, managing finances) are normally used to assess and monitor cognitive functions and the ability to live independently.  ADLS (feeding, dressing, bathing and walking) are instead predictors of need of admission to nursing homes, alternative living arrangements, hospitalization or use of home care. In your paper, you mix up different Instrumental and Basic activities (you include walking in Instrumental) and relate these to a generic self-reported (biased) health status of elderly people. You should consider that there are, for instance, at international level, several on-going researches on mental health and outdoor environments (mental health is very different from a generic 'health' status). This confusion of terms, methods and approach is a major limit of this study, and I suggest you use a more specific term and, as an example, avoid referring to elderly (generic) 'health' as such.

Line 134 Add a space after “al.”.

Lines 157, 244, 252, 253 Change “Residence” with “Residential area”.

Line 167 Add a space after “al.”.

Lines 203-204 This is another major limit of this study, you totally ignore that behavioral gender differences are often determined by country-specific “Cultural aspects” (e.g. in my country no man is used to do laundry, and elderly male people are always more likely to participate in individual activities, whereas elderly female people are always more likely to take part in social activities).

Lines 322-329 It is not clear to me, consequently, how the findings of this study are “helpful for developing strategies to improve elderly people’s health…”. Results are limited and controversial and often provide simplistic 'strategies'. You should consider that major international research initiatives are focusing scientific efforts  investigating “types and characteristics of outdoor environments and their relations to very specific health problems (e.g. mental) in different population categories”. Moreover, you discuss about the living “Environment”, when in your recommendations for "developing strategies to improve elderly people’s health” you refer to the possible organization of different gender-related activities, such as “community lectures, interest groups, or voluntary activities”  for women (Line 325), which can be potentially be carried out in any kind of environment (indoor, outdoor, etc.)

Resolution quality of all images must be improved.

Author Response

Response to Review 1 Comments

Lines 50-51 "It is easy to intervene....": In what sense it is easy? I believe it not easy at all!

Response:Apologize for the inaccuracy. Thanks very much for the comment, and the sentence is changed into “it is possible”, as detailed in line 49.

Lines 53-54 Repetition, please erase.

Response:Thanks very much for the comment, but apologize that we could not find the place where is repeated.

Line 66 The "few studies" must be cited.

Response:In fact, we did not find any literature on it, but to avoid being too categorical, we use "few studies"。

Line 72 The "theory of social ecology" must be cited.

Response:References have been added, as detailed in line 64.

Line 81 The "previous social ecology studies" must be cited.

Response:References have been added, as detailed in line 73.

Line 91 The "Social Ecology framework" must be cited.

Response:References have been added, as detailed in line 83.

Line 93 "Living environment" is a vague term, do you mean "Outdoor environment"? There are several scientific studies concentrating on Indoor living environment and human health.

Response:Thanks very much for your comments. Our research is indeed about the outdoor environment in the community, therefore, we change "Living environment" in the manuscript into "Outdoor environment".

Lines 94-96 Bold statements on own research should be avoided.

Response:Thanks very much for your comments. We delete the statement: The conclusions will not only help us to understand the enigma of the gender differences in the elderly people’s environment, behavior, and health but they will also improve the health of both genders and enhance the health of older people overall.

Line 124 Erase "and so on".

Response:"and so on" has been deleted, as detailed in line 159.

Lines 127-128 The sentence (and possibly the whole study, with this respect) is not clear, please explain better. Please note that IALS (i.e. cooking, cleaning, transportation, laundry, managing finances) are normally used to assess and monitor cognitive functions and the ability to live independently.  ADLS (feeding, dressing, bathing and walking) are instead predictors of need of admission to nursing homes, alternative living arrangements, hospitalization or use of home care. In your paper, you mix up different Instrumental and Basic activities (you include walking in Instrumental) and relate these to a generic self-reported (biased) health status of elderly people. You should consider that there are, for instance, at international level, several on-going researches on mental health and outdoor environments (mental health is very different from a generic 'health' status). This confusion of terms, methods and approach is a major limit of this study, and I suggest you use a more specific term and, as an example, avoid referring to elderly (generic) 'health' as such.

Response:Activity ability refers to the basic and common activities that people carry out repeatedly every day in order to survive and adapt to the living environment, and it is usually composed of two parts, ADL (Activity of Daily Living) and IADL (Instrumental Activity of Daily Living). ADL focuses on whether elderly people have functional disability, including dressing, showering, eating, getting out of bed, going to toilet, and controlling the flow of urine and feces, while IDAL tends to reflect the ability of the elderly to living independently and adapt to social and family roles.

The research paid attention on the relationship between the environment, elderly behavior and their health, and the elderly people who did not have functional disability are studied. Therefore, the survey conducted by Fudan University mainly focused on IADL of elderly people, in terms of scholar Lawton‘s instrumental activities of daily living scale, which was the most widely used (Lawton & Brody, 1969; Mattos et al, 2014). The scale contains 8 indicators, including shopping, outdoor exercise, cooking food, household chores, washing clothes, using a telephone, taking medications, and handling finances, and we kept 7 indicators of them except taking medications, because it failed the confirmatory factor analysis.

After the revision, the paper added descriptive statistics of variables and sample situations, as detailed in line 235-247.

  1. Lawton, M. P.; Brody, E. M. Assessment of older people: self-maintaining and instrumental activities of daily living. Gerontologist [Online] 1969, 9 (3), 179.
  2. Mattos, I. E.; Do Carmo, C. N.; Santiago, L. M.; Luz, L. L. Factors associated with functional incapacity in elders living in long stay institutions in Brazil: a cross-sectional study. BMC Geriatrics [Online] 2014, 14 (1), 47. https://dx.doi.org/10.1186/1471-2318-14-47.

In addition, some “health” in the paper has been changed into “activity ability”, which is supposed to be more specific.

Line 134 Add a space after “al.”.

Response:A space after “al.” is added, as detailed in line 176.

Lines 157, 244, 252, 253 Change “Residence” with “Residential area”.

Response: We change “residence” into “residential area”, as detailed in line 199, 204, 315.

Line 167 Add a space after “al.”.

Response:A space after “al.” is added, as detailed in line 213.

Lines 203-204 This is another major limit of this study, you totally ignore that behavioral gender differences are often determined by country-specific “Cultural aspects” (e.g. in my country no man is used to do laundry, and elderly male people are always more likely to participate in individual activities, whereas elderly female people are always more likely to take part in social activities).

Response:Indeed, “in my country no man is used to do laundry, and elderly male people are always more likely to participate in individual activities, whereas elderly female people are always more likely to take part in social activities”, this feature does not conflict with our research, and it is confirmed by our research as well. The research is more concerned with the gender differences in how outdoor environments affect the activity ability of the elderly, and we found that, elderly males were more easily influenced by the objective environment, while elderly females were more influenced by social activities.

Besides, according to the confirmatory factor analysis of the IADL measurement model, the factor load of each measurement variable was more than 0.6, which meant that the model that was based on the survey had good validity and reliability, although there were some gender differences in the habits. As far as individuals, the latent IADL could represent the activity ability effectively.

Lines 322-329 It is not clear to me, consequently, how the findings of this study are “helpful for developing strategies to improve elderly people’s health…”. Results are limited and controversial and often provide simplistic 'strategies'. You should consider that major international research initiatives are focusing scientific efforts  investigating “types and characteristics of outdoor environments and their relations to very specific health problems (e.g. mental) in different population categories”. Moreover, you discuss about the living “Environment”, when in your recommendations for "developing strategies to improve elderly people’s health” you refer to the possible organization of different gender-related activities, such as “community lectures, interest groups, or voluntary activities”  for women (Line 325), which can be potentially be carried out in any kind of environment (indoor, outdoor, etc.)

Response:(1) Thanks very much of the comments. The study discussed the gender differences in the relationship between outdoor environment, elderly behavior and their activity ability. We found that the activity ability of the male elderly was more affected by the objective outdoor environment and walking environment, while that of the female elderly was more affected by neighborhood association and social participation. Therefore, taking the gender differences into account, specific care and design can be made to improve the aging-friendly community environment and governance, for example, the characteristics of male elderly should be considered more in the design of walking environment, and more community activities are to be organized specifically for the female elderly. This will help to improve the activity ability of older men and women, respectively.

(2) There are indeed many research on “types and characteristics of outdoor environments and their relations to very specific health problems (e.g. mental) in different population categories, but with the development of Post Occupation Evaluation (POE) theory, subjective assessment of the built environment has become one of the main methods of environment assessment in recent years (Lin & Moudon, 2010; Sayegh et al., 2016), which is a real response of the users to the living environment. It can not only evaluate the quality of the existing environment, but also be used as a feedback mechanism to improve the environment. It also helps to promote the theory and practice development of urban planning, environment and architectural design. In the original manuscript, there is not much discussion on the subject evaluation of the built environment, which has been added in the revision, as detailed in line 168-174.

  1. Lin L , Moudon A V . Objective versus subjective measures of the built environment, which are most effective in capturing associations with walking?[J]. Health & Place, 2010, 16(2):339-348.
  2. Sayegh A , Andreani S , Kapelonis C , et al. Experiencing the built environment: strategies to measure objective and subjective qualities of places[J]. Open Geospatial Data Software & Standards, 2016, 1(1):11.

Further study will be made combining the subjective and objective assessment, which is explained in the research limitations and follow-ups in the final part of the paper, as detailed in line 414-418.

Resolution quality of all images must be improved.

Response:The color of the figures are changed, and their resolution is improved.

Reviewer 2 Report

The article “The enigma of gender differences in an Environment-Behavior- Health model of elderly people’s health: The choice between individuality and sociality” investigates gender difference in the health of elderly people. The article is interesting but needs to be improved in several aspects which are listed below.

  • The abstract must be reviewed as the methodological aspects are completely missing.
  • A paragraph must be provided for the presentation of the hypotheses. The relations shown in figure 3 must be better detailed, showing the signs (positive or negative) of these relations.
  • A paragraph is missing that presents the socio-demographic characteristics of the study participants
  • Cronbach's alpha of the measures used must be reported
  • The analyzes carried out for each formulated hypothesis must be specified
  • In presenting the results, recall the hypotheses formulated
  • The fit indices of the SEM conducted are not reported
  • Among the limitations of the study it should also be reported that the data were collected in 2014. Therefore further data collections should be carried out.

Author Response

Response to Review 2 Comments

  • The abstract must be reviewed as the methodological aspects are completely missing.
  • Response:A discussion of methodology has been added in the abstract, as detailed in line 14-19.
  • A paragraph must be provided for the presentation of the hypotheses.
  • Response:We appreciate the comments very much, and a section about research hypothesis is added after Introduction, as detailed in line 89-129.
  • The relations shown in figure 3 must be better detailed, showing the signs (positive or negative) of these relations.
  • Response:Figure 3 shows the comparison based on the mean of variables between the male and female samples, therefore, there is no such signs (positive or negative).
  • A paragraph is missing that presents the socio-demographic characteristics of the study participants
  • Response:The descriptive statistics of the variables have been added in the section of Results, as detailed in line 235-247.
  • Cronbach's alpha of the measures used must be reported
  • Response:The scores of Cronbach’s alpha are explained in line 223-225.
  • The analyzes carried out for each formulated hypothesis must be specified
  • Response:The validations of the Hypotheses have been added in the Results, as detailed in line 282, 283, 298, 299, 307, 308, 325,326. Besides, the comparison table 5 is added in line 327.
  • In presenting the results, recall the hypotheses formulated
  • Response:In section 4.3, the validations of the Hypotheses have been added, and the discussion of this part is elaborated around the Hypotheses, as detailed in line 272-328.
  • The fit indices of the SEM conducted are not reported
  • Response:The fit indexes of the SEM have been added, as detailed in line 226-232.
  • Among the limitations of the study it should also be reported that the data were collected in 2014. Therefore further data collections should be carried out.
  • Response:We appreciate the comments very much, and indeed, the data we used is based on a survey done in 2014, which has been a long time, and that is one limitation of our paper. We are currently conducting further research on the environment and people’s health in this community, and we are going to get more updated data. This limitation has been also added in the last paragraph of the discussion part of the paper, as detailed in line 414-418.

Reviewer 3 Report

An interesting and well-written manuscript. Some details:

Page 5 line 197 and Page 7 line 218. A parentheses is lacking around P-values. 

There are inconsistencies in the use of upper case and lowercase in journals (e.g. 2, 6, 8, 10, 14) and in the titles (7, 8, 11, 12, 14, 16, 17, 23, 27, 28, 29, 30, 36, 37, 39,) in the reference list.

Author Response

Response to Review 3 Comments

An interesting and well-written manuscript. Some details:

Page 5 line 197 and Page 7 line 218. A parentheses is lacking around P-values. 

Response:The parentheses around P-values has been added,as detailed in line 260, 278. 

There are inconsistencies in the use of upper case and lowercase in journals (e.g. 2, 6, 8, 10, 14) and in the titles (7, 8, 11, 12, 14, 16, 17, 23, 27, 28, 29, 30, 36, 37, 39,) in the reference list.

 Response:We appreciate the suggestions, and the reference has been modified.

Round 2

Reviewer 1 Report

I suggest to change the Title, currently redundant: "....model for elderly people:..." (i.e. avoid repeating health).

Abstract: I suggest changing the final sentente into: " the study provided some useful insights for future research, aiming at  promoting gender-sensitive urban design."

Move "scope for further research fro Discussion to Conclusions, also because concluding remarks are too short. Limits of the conducted study should be included in the Conclusions as well.

Line 16: 2,839. Erase 'of elderly people' (repetition

Figure 3: source is missing

Line 19: with the social ecology theory

Line 168: POE should be referenced, both in text and the appropriate section

Line 420: 'explored' (not 'discovered')

Line 425: the outdoor environment

Lines 425-426: 'the different genders' uses' (i.e. avoid repetition)

Author Response

I suggest to change the Title, currently redundant: "....model for elderly people:..." (i.e. avoid repeating health).

Response: Thanks for the comments, and the second “health” is erased, as detailed in line 3.

Abstract: I suggest changing the final sentente into: " the study provided some useful insights for future research, aiming at promoting gender-sensitive urban design."

Response: Thanks for the comments, and this sentence has been changed, as detailed in line 24-25.

Move "scope for further research fro Discussion to Conclusions, also because concluding remarks are too short. Limits of the conducted study should be included in the Conclusions as well.

Response: Thanks for the comments, and the limitations and further research have been moved to the Conclusion, as detailed in line 413-427.

Line 16: 2,839. Erase 'of elderly people' (repetition

Response: Thanks for the comments, and “of elderly people” is erased, as detailed in line 16.

Figure 3: source is missing

Response: “Source: based on google map, drawn by the author” has been added, as detailed in line 155.

Line 19: with the social ecology theory

Response: Thanks for the comments, and “the” is added, as detailed in line 19.

Line 168: POE should be referenced, both in text and the appropriate section

Response: Thanks for the comments, and refence [42] is added, as detailed in line 168.

Line 420: 'explored' (not 'discovered')

Response: Thanks for the comments, and “explored” is used, as detailed in line 405.

Line 425: the outdoor environment

Response: Thanks for the comments, and “outdoor” is added, as detailed in line 410.

Lines 425-426: 'the different genders' uses' (i.e. avoid repetition)

Response: Thanks for the comments, and sentence has been changed, as detailed in line 411.